# An Unbiased Risk Estimator for Learning with Augmented Classes

**Yu-Jie Zhang, Peng Zhao, Lanjihong Ma, Zhi-Hua Zhou**
National Key Laboratory for Novel Software Technology,
Nanjing University, Nanjing 210023, China
{zhangyj, zhaop, maljh, zhouzh}@lamda.nju.edu.cn

## Abstract

This paper studies the problem of learning with augmented classes (LAC), where augmented classes unobserved in the training data might emerge in the testing phase. Previous studies generally attempt to discover augmented classes by exploiting geometric properties, achieving inspiring empirical performance yet lacking theoretical understandings particularly on the generalization ability. In this paper we show that, by using unlabeled training data to approximate the potential distribution of augmented classes, an unbiased risk estimator of the testing distribution can be established for the LAC problem under mild assumptions, which paves a way to develop a sound approach with theoretical guarantees. Moreover, the proposed approach can adapt to complex changing environments where augmented classes may appear and the prior of known classes may change simultaneously. Extensive experiments confirm the effectiveness of our proposed approach.

## 1 Introduction

Recent advances in machine learning encourage its application in high-stake scenarios, where the robustness is the central requirement [1, 2]. A robust learning system should be able to handle the distribution change in the non-stationary environments [3, 4, 5]. In this paper, we focus on the problem of learning with augmented classes (LAC) [6], where the class distribution changes during the learning process—some augmented classes unobserved in training data might emerge in testing. To make reliable predictions, desired learning systems are required to identify augmented classes and retain good generalization performance over the testing distribution.

The main challenge of the LAC problem lies in how to depict relationships between known and augmented classes. A typical solution is to learn a compact geometric description of the known classes and take those beyond the description as augmented classes, where the anomaly detection or novelty detection approaches can be employed (such as one-class SVM [7, 8], kernel density estimation [9, 10] and iForest [11]). Da et al. [6] give the name of LAC and employ the low-density separation assumption to adjust the decision boundaries in a multi-class situation. In addition to the effort of machine learning community, the computer vision and pattern recognition communities also contribute to the study of the problem (or its cousin). Scheirer et al. [12] propose the notion of open space risk to penalize predictions outside the support of training data, based on which several approaches are developed [12, 13]. Later, approaches based on the nearest neighbor [14] and extreme value theory [15] are also developed. More discussions on related topics are deferred to Section 5.

Although various approaches are proposed with nice performance and some of them conduct theoretical analysis, generalization properties of the LAC problem is less explored. Scheirer et al. [12, 13], Rudd et al. [15] formally use the open space risk or extreme value theory to identify augmented classes, but the generalization error of learned models is not further analyzed. There are also works [16, 17, 18] focusing on the Neyman-Pearson (NP) classification, which controls the novelty

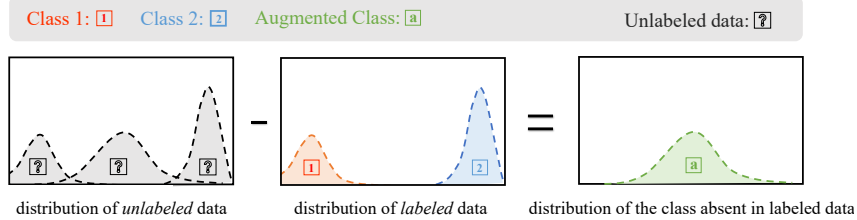

Figure 1: Distribution of augmented classes can be estimated by those of labeled and unlabeled training data.

detection ratio of augmented classes or false positive ratio of known classes with the constraint on another. By using unlabeled data, authors develop approaches with one-side PAC-style guarantees for the binary NP classification whereas the generalization ability for the LAC problem is not studied.

To design approaches with generalization error guarantees for the LAC problem, it is necessary to assess the distribution of augmented classes in the training stage. Note that in many applications, during the training stage, in addition to labeled data, there are abundant unlabeled training data available. In this paper, we show that by exploiting unlabeled training data, an *unbiased* risk estimator over the *testing* distribution can be established under mild assumptions. The intuition is that, though instances from augmented classes are unobserved from labeled data, their distribution information may be contained in unlabeled data and estimated by separating the distribution of known classes from unlabeled data (Figure 1). More concretely, we propose the *class shift condition* to model the testing distribution as a mixture of known and augmented classes' distributions. Under such a condition, classifiers' risk over testing distribution can be estimated in the training stage, where minimizing its empirical estimator finally gives our EULAC approach, short for Exploiting Unlabeled data for Learning with Augmented Classes. Moreover, the EULAC approach can further take the prior change on known classes into account, which enables its adaptivity to complex changing environments.

EULAC enjoys several favorable properties. Theoretically, our approach has both asymptotic (consistency) and non-asymptotic (generalization error bound) guarantees. Notably, the non-asymptotic analysis further justifies the capability of our approach in exploiting unlabeled data, since the generalization error becomes smaller with an increasing number of unlabeled data. Moreover, extensive experiments validate the effectiveness of our approach. It is noteworthy to mention that our approach can now perform the standard cross validation procedure to select parameters, while most geometric-based approaches cannot due to the unavailability of the testing distribution, and their parameters setting heavily relies on the experience. We summarize main contributions of this paper as follows.

(1) We propose the *class shift condition* to characterize the connection between known and augmented classes for the learning with augmented class problem.
(2) Based on the class shift condition, we establish an *unbiased risk estimator* over the testing distribution for the LAC problem by exploiting the unlabeled data. Similar results are also attainable for a general setting of class distribution change.
(3) We develop our EULAC approach with the unbiased risk estimator, whose theoretical effectiveness is proved by both consistency and generalization error analyses. We also conduct extensive experiments to validate its empirical superiority.

## 2 An Unbiased Risk Estimator for LAC problem

In this section, we formally describe the LAC problem, followed by the introduction of the class shift condition, based on which we develop the unbiased risk estimator over the testing distribution. Moreover, we show the potential of our approach for adapting to complex changing environments.

### 2.1 Problem Setup and Class Shift Condition

**LAC problem.** In the training stage, the learner collects a labeled dataset $D_L = \{(\mathbf{x}_i, y_i)\}_{i=1}^{n_l}$ sampled from distribution of known classes $P_{\mathsf{kc}}$ defined over $\mathcal{X} \times \mathcal{Y}'$, where $\mathcal{X}$ denotes the feature space and $\mathcal{Y}' = \{1, \ldots, K\}$ is the label space of $K$ known classes. In the testing stage, the learner requires to predict instances from the testing distribution $P_{te}$, where augmented classes not observed before might emerge. Since the specific partition of augmented classes is unknown, the learner will predict all of them as a single augmented class $\mathsf{ac}$. So the testing distribution is defined over

$\mathcal{X} \times \mathcal{Y}$, where $\mathcal{Y} = \{1, \ldots, K, \mathsf{ac}\}$ is the augmented label space. The goal of the learner is to train a classifier $f : \mathcal{X} \mapsto \mathcal{Y}$ achieving good generalization ability by minimizing the expected risk $R(f) = \mathbb{E}_{(\mathbf{x},y) \sim P_{te}} [\mathbb{1}(f(\mathbf{x}) \neq y)]$ over the *testing* distribution, where $\mathbb{1}(\cdot)$ is the indicator function.

**Unlabeled data.** In our setup, the learner additionally receives a set of *unlabeled data* $D_U = \{\mathbf{x}_i\}_{i=1}^{n_u}$ sampled from the testing distribution and hopes to use it to enhance performance of the trained classifier. This learning scenario happens when labeled training data fail to capture certain classes of the testing distribution due to the class distribution change, while we can easily collect a vast amount of unlabeled data from current environments. Essentially, the missed class information has already been contained in the training data (unlabeled data) though is not revealed in the supervision (labeled training data). We thus prefer to call such classes as the "augmented class" instead of "new class".

**Class shift condition.** Although not explicitly stated, previous works [12, 6, 14] essentially rely on the assumption that the distribution of known classes remains unchanged when augmented classes emerge. Following the same spirit, we introduce the following *class shift condition* for the LAC problem to rigorously depict the connection between known and augmented class distributions.

**Definition 1** (Class Shift Condition). The testing distribution $P_{te}$, the distribution of known classes $P_{\mathsf{kc}}$ and the distribution of augmented classes $P_{\mathsf{ac}}$ are under the *class shift condition*, if

$$P_{te} = \theta \cdot P_{\mathsf{kc}} + (1 - \theta) \cdot P_{\mathsf{ac}}, \tag{1}$$

where $\theta \in [0, 1]$ is a certain mixture proportion.[1]

Class shift condition states that the testing distribution can be regarded as a mixture of those of known and augmented classes with a certain proportion $\theta$, based on which we can evaluate classifiers' risk over the testing distribution with labeled and unlabeled training data.

## 2.2 Convex Unbiased Risk Estimator

This part, we develop an unbiased risk estimator for the LAC problem under the class shift condition. We first introduce the notation conventions. The density function is denoted by the lowercase $p$, and the joint, conditional and marginal density functions are indicated by the subscripts $XY$, $X|Y$ ($Y|X$) and $X$ ($Y$). For instance, $p_X^{te}(\mathbf{x})$ refers to the marginal density of the testing distribution over $\mathcal{X}$.

**OVR scheme.** Suppose the joint testing distribution were available, the LAC problem would degenerate to standard multi-class classification, which can be then addressed by existing approaches. Among those approaches, we adopt the one-versus-rest (OVR) strategy, which enjoys sound theoretical guarantees [19] and nice practical performance [20]. The risk minimization is formulated as,

$$\min_{f_1,\ldots,f_{K+1}} R_\psi(f_1, \ldots, f_{K+1}) = \mathbb{E}_{(\mathbf{x},y) \sim P_{te}} \left[ \psi(f_y(\mathbf{x})) + \sum_{k=1, k \neq y}^{K+1} \psi(-f_k(\mathbf{x})) \right], \tag{2}$$

where $f_k$ is the classifier for the $k$-th class, $k = 1, \ldots, K$; and $f_{\mathsf{ac}}$ is the classifier for the augmented class. For simplicity, we substitute $f_{\mathsf{ac}}$ with $f_{K+1}$ in the formulation. $\psi : \mathbb{R} \mapsto [0, +\infty)$ is a binary surrogate loss such as hinge loss. The OVR scheme predicts by $f(\mathbf{x}) = \arg\max_{k \in \{1,\ldots,K,\mathsf{ac}\}} f_k(\mathbf{x})$.

**Approximating the testing distribution.** However, *the joint testing distribution is unavailable in the training stage due to the absence of labeled instances from augmented classes.* Fortunately, we show that given the mixture proportion $\theta$, it can be approximated with the labeled and unlabeled data. Under the class shift condition, the joint density of the testing distribution can be decomposed as

$$
\begin{aligned}
p_{XY}^{te}(\mathbf{x}, y) &\overset{(1)}{=} \theta \cdot p_{XY}^{\mathsf{kc}}(\mathbf{x}, y) + (1 - \theta) \cdot p_{XY}^{\mathsf{ac}}(\mathbf{x}, y) \\
&= \theta \cdot p_{XY}^{\mathsf{kc}}(\mathbf{x}, y) + \mathbb{1}(y = \mathsf{ac}) \cdot (1 - \theta) \cdot p_X^{\mathsf{ac}}(\mathbf{x}),
\end{aligned} \tag{3}
$$

where the last equality follows from the fact that $p_{XY}^{\mathsf{ac}}(\mathbf{x}, y) = 0$ holds for all $\mathbf{x} \in \mathcal{X}$ and $y \neq \mathsf{ac}$. The first part $p_{XY}^{\mathsf{kc}}(\mathbf{x}, y)$ is accessible via the labeled data. The only unknown term is the second part, the marginal density of the augmented class $p_X^{\mathsf{ac}}(\mathbf{x})$. Under the class shift condition, it can be evaluated by separating the distribution of labeled data from unlabeled data as

$$(1 - \theta) \cdot p_X^{\mathsf{ac}}(\mathbf{x}) = p_X^{te}(\mathbf{x}) - \theta \cdot p_X^{\mathsf{kc}}(\mathbf{x}). \tag{4}$$

Thus, by plugging (4) into (3), the testing distribution becomes attainable, and consequently, we can evaluate the OVR risk $R_\psi$ in the training stage through an equivalent risk $R_{LAC}$.

**Proposition 1.** *Under the class shift condition, for measurable functions $f_1, \ldots, f_K, f_{\mathsf{ac}}$, we have $R_\psi(f_1, \ldots, f_K, f_{\mathsf{ac}}) = R_{LAC}(f_1, \ldots, f_K, f_{\mathsf{ac}})$, where $R_{LAC}$ is defined as,*

$$R_{LAC} = \theta \cdot \mathbb{E}_{(\mathbf{x},y) \sim P_{\mathsf{kc}}} \left[ \psi(f_y(\mathbf{x})) - \psi(-f_y(\mathbf{x})) + \psi(-f_{\mathsf{ac}}(\mathbf{x})) - \psi(f_{\mathsf{ac}}(\mathbf{x})) \right]$$

$$+ \mathbb{E}_{\mathbf{x} \sim p_X^{te}(\mathbf{x})} \left[ \psi(f_{\mathsf{ac}}(\mathbf{x})) + \sum_{k=1}^{K} \psi(-f_k(\mathbf{x})) \right]. \tag{5}$$

**Remark 1.** We can assess $R_{LAC}$ during training as the distribution of known classes $P_{\mathsf{kc}}$ and marginal testing distribution $p_X^{te}(\mathbf{x})$ can be estimated by labeled and unlabeled training data, respectively.

The remaining issue for the LAC risk $R_{LAC}$ is the non-convexity caused by terms $-\psi(-f_y(\mathbf{x}))$ and $-\psi(f_{\mathsf{ac}}(\mathbf{x}))$, which are non-convex w.r.t the classifiers even with the convex binary surrogate loss $\psi$. Inspired by studies [21, 22], we can eliminate the non-convexity by carefully choosing the surrogate loss satisfying $\psi(z) - \psi(-z) = -z$ for all $z \in \mathbb{R}$, and thereby $R_{LAC}$ enjoys a convex formulation

$$R_{LAC} = \theta \cdot \mathbb{E}_{(\mathbf{x},y) \sim P_{\mathsf{kc}}} \left[ f_{\mathsf{ac}}(\mathbf{x}) - f_y(\mathbf{x}) \right] + \mathbb{E}_{\mathbf{x} \sim p_X^{te}(\mathbf{x})} \left[ \psi(f_{\mathsf{ac}}(\mathbf{x})) + \sum_{k=1}^{K} \psi(-f_k(\mathbf{x})) \right]. \tag{6}$$

Many loss functions satisfy the above condition [22], such as logistic loss $\psi(z) = \log(1 + \exp(-z))$, square loss $\psi(z) = (1-z)^2/4$ and double hinge loss $\psi(z) = \max(-z, \max(0, (1-z)/2))$. Since LAC risk $R_{LAC}$ equals to the ideal OVR risk $R_\psi$, its empirical estimator $\widehat{R}_{LAC}$ is *unbiased* over the testing distribution. We can thus perform the standard empirical risk minimization. Finally, we note that Proposition 1 can be generalized for arbitrary multiclass losses, if the convexity is not required, where more multiclass and binary losses can be used. We will take this as a future work.

## 2.3 Convex Unbiased Risk Estimator under Generalized Class Shift Condition

The class shift condition in Definition 1 models the appearance of augmented classes with the assumption that the distribution of known classes is identical to that in the testing stage. In real-world applications, however, the environments might be more complex, where the distribution of known classes could also shift. We consider a specific kind of class distribution change: in addition to the emerging augmented classes, the prior of each class $p_Y^{te}(y)$ varies from labeled data to testing data, while their conditional density remains the same, namely $p_{X|Y}^{te}(\mathbf{x}|y) = p_{X|Y}^{\mathsf{kc}}(\mathbf{x}|y)$ for all $y \in [K]$. To this end, we propose following *generalized class shift condition* to model such a case by further decomposing the distribution of known classes in the testing stage as a mixture of several components,

$$P_{te} = \sum_{k=1}^{K} \theta_{te}^k \cdot P_k + \left( 1 - \sum_{k=1}^{K} \theta_{te}^k \right) \cdot P_{\mathsf{ac}}, \tag{7}$$

where $P_k$ is the distribution of the $k$-th known class whose marginal density equals to $p_{X|Y}^{\mathsf{kc}}(\mathbf{x}|k)$, and $\theta_{te}^k = p_Y^{te}(k)$ is the prior of $k$-th known class in testing, for all $k \in [K]$. When there is no distribution change on known classes, the generalized class shift condition recovers the vanilla version in (1).

With the generalized class shift condition (7), following the similar argument in Section 2.2, we can evaluate the OVR risk for the testing distribution even if the prior of known classes has changed.

**Proposition 2.** *Under the generalized class shift condition (7), by choosing the surrogate loss function satisfying $\psi(z) - \psi(z) = -z$ for all $z \in \mathbb{R}$, for measurable functions $f_1, \ldots, f_K, f_{\mathsf{ac}}$, we have $R_\psi(f_1, \ldots, f_K, f_{\mathsf{ac}}) = R_{LAC}^{shift}(f_1, \ldots, f_K, f_{\mathsf{ac}})$, where $R_{LAC}^{shift}$ is defined as,*

$$R_{LAC}^{shift} = \sum_{k=1}^{K} \theta_{te}^k \cdot \mathbb{E}_{(\mathbf{x},y) \sim P_k} \left[ f_{\mathsf{ac}}(\mathbf{x}) - f_y(\mathbf{x}) \right] + \mathbb{E}_{\mathbf{x} \sim p_X^{te}(\mathbf{x})} \left[ \psi(f_{\mathsf{ac}}(\mathbf{x})) + \sum_{k=1}^{K} \psi(-f_k(\mathbf{x})) \right].$$

Proposition 2 implies that we can handle the augmented classes together with the distribution change on prior of known classes by empirically minimizing the risk $R_{LAC}^{shift}$. Note that since $R_{LAC}^{shift}$ further decomposes the distribution of known classes into several components, it enjoys more flexibility than $R_{LAC}$ in evaluating the testing risk, yet requires more efforts in estimation of class prior $\theta_{te}^k$ for each known class rather than mixture proportion $\theta$ only, which will be discussed next.

## 3 Approach

In this section, we develop two practical algorithms for the proposed EULAC approach to minimize the empirical version of the LAC risk $R_{LAC}$ (similar results can be extended for its generalization $R_{LAC}^{shift}$). Meanwhile, we discuss how to estimate the mixture proportion $\theta$ and class prior $\theta_{te}^k$.

**Kernel-based hypothesis space.** We first consider minimizing the empirical LAC risk $\widehat{R}_{LAC}$ in the reproducing kernel Hilbert space (RKHS) $\mathbb{F}$ associated to a PDS kernel $\kappa : \mathcal{X} \times \mathcal{X} \mapsto \mathbb{R}$:

$$\min_{f_1,\ldots,f_K,f_{\text{ac}}\in\mathbb{F}} \widehat{R}_{LAC} + \lambda\Big( \sum\nolimits_{k=1}^{K} \|f_k\|_{\mathbb{F}}^2 + \|f_{\text{ac}}\|_{\mathbb{F}}^2 \Big), \tag{8}$$

where $\widehat{R}_{LAC}$ is the empirical approximation of the LAC risk (6)

$$\widehat{R}_{LAC} = \frac{\theta}{n_l} \sum\nolimits_{i=1}^{n_l} (f_{\text{ac}}(\mathbf{x}_i) - f_{y_i}(\mathbf{x}_i)) + \frac{1}{n_u} \sum\nolimits_{i=1}^{n_u} \Big( \psi(f_{\text{ac}}(\mathbf{x}_i)) + \sum\nolimits_{k=1}^{K} \psi(-f_k(\mathbf{x}_i)) \Big). \tag{9}$$

According to the representer theorem [23], the optimal solution of (8) is provably in the form of

$$f_k(\cdot) = \sum\nolimits_{\mathbf{x}_i \in D_L} \alpha_i^k \kappa(\cdot, \mathbf{x}_i) + \sum\nolimits_{x_j \in D_U} \alpha_j^k \kappa(\cdot, \mathbf{x}_j), \tag{10}$$

where $\alpha_i^k$ is the $i$-th coefficient of the $k$-th classifier. Plugging (10) into (8), we get a convex optimization problem with respect to $\boldsymbol{\alpha}$, which can be solved efficiently. Since the risk estimator $\widehat{R}_{LAC}$ is assessed on the testing distribution directly, we can perform *unbiased* cross validation to select parameters. Then, after obtaining the binary classifiers $f_1, \ldots, f_K, f_{\text{ac}}$, we follow the OVR rule to construct the final predictor as $f : \mathcal{X} \mapsto \mathcal{Y}$ with $f(\mathbf{x}) = \arg\max_{k\in\{1,\ldots,K,\text{ac}\}} f_k(\mathbf{x})$.

**Deep model.** Our approach can be also implemented by deep neural networks. Since the deep models themselves are non-convex, we directly minimize the non-convex formulation of $R_{LAC}$ (5) by taking outputs of the deep model as OVR classifiers. However, as shown by Kiryo et al. [24], the direct minimization easily suffers from over-fitting as the risk is not bounded from below by 0. To avoid the undesired phenomenon, we apply their proposed non-negative risk [24] to rectify the OVR scheme for training the deep model, whose effectiveness will be validated by experiments. More detailed elaborations for the rectified $R_{LAC}$ risk is presented in the full paper [25].

**On the estimation of $\theta$.** Notice that minimizing $\widehat{R}_{LAC}$ requires estimating $\theta$, which is known as the problem of *Mixture Proportion Estimation* (MPE) [26], where one aims to estimate the maximum proportion of distribution $H$ in distribution $F$ given their empirical observations. Many works have been devoted to developing theoretical foundations and efficient algorithms [27, 17, 28, 29]. We employ the kernel mean embedding (KME) based algorithm proposed by Ramaswamy et al. [26], which guarantees that the estimator $\widehat{\theta}$ converges to true proportion $\theta$ in the rate of $\mathcal{O}(1/\sqrt{\min\{n_l, n_u\}})$ under the separability condition. Moreover, since the KME-based algorithm easily suffers from the curse of dimensionality in practice, inspired by the recent work [28], we further use a pre-trained model to reduce the dimensionality of original input to its probability outputs. We refer to the above estimator as KME-base, and the corresponding approach for LAC as EULAC-base.

Additionally, under the generalized class shift condition, we need more refined estimations for each known class. Therefore, we use the above MPE estimator to estimate each class prior $\theta_{te}^k$ in $R_{LAC}^{shift}$ (2) via the labeled instances from the $k$-th known class and the unlabeled data, $k \in [K]$. We refer to such an estimator as KME-shift and the corresponding approach as EULAC-shift. Finally, we note that since the vanilla LAC can also be modeled with the generalized class shift condition, we can use KME-shift to estimate the mixture proportion $\widehat{\theta}$ by $\widehat{\theta} = \sum_{k=1}^{K} \widehat{\theta}_{te}^k$. It turns out that KME-shift achieves comparable (even better) empirical performance with KME-base.

# 4 Theoretical Analysis

In this section, we first show the infinite-sample *consistency* of the LAC risk $R_{LAC}$. Then, we derive the *generalization error* bounds. All the proofs can be found in the full paper [25].

**Infinite-sample consistency.** At first, we show that the LAC risk $R_{LAC}$ is infinite-sample consistent with the risk over the testing distribution with respect to 0-1 loss. Namely, by minimizing the expected risk of $R_{LAC}$, we can get classifiers achieving the Bayes rule over the testing distribution.

**Theorem 1.** *Under the class shift condition, suppose the surrogate loss $\psi$ is convex, bounded below, differential, satisfying $\psi(z) - \psi(-z) = -z$ and $\psi(z) < \psi(-z)$ when $z > 0$, then for any $\epsilon_1 > 0$, there exists $\epsilon_2 > 0$ such that*

$$R_{LAC}(f_1, \ldots, f_K, f_{\text{ac}}) \leq R_{LAC}^* + \epsilon_2 \quad \Longrightarrow \quad R(f) \leq R^* + \epsilon_1$$

holds for all measurable functions $f_1, \ldots, f_K, f_{\mathsf{ac}}$ and $f(\mathbf{x}) = \arg\max_{k \in \{1, \ldots, K, \mathsf{ac}\}} f_k(\mathbf{x})$. Here, $R^*_{LAC} = \min_{f_1, \ldots, f_K, f_{\mathsf{ac}}} R_{LAC}(f_1, \ldots, f_K, f_{\mathsf{ac}})$ and $R^* = \min_f R(f) = \mathbb{E}_{(\mathbf{x}, y) \sim P_{te}} [\mathbb{1}(f(\mathbf{x}) \neq y)]$ is the Bayes error over the testing distribution.

Theorem 1 follows from Proposition 1 and analysis in the seminal work of Zhang [19], who investigates the consistency property of OVR risk in depth. Since the LAC risk $R_{LAC}$ is equivalent to the OVR risk $R_\psi$, it is naturally infinite-sample consistent. There are many loss functions satisfy assumptions in Theorem 1 such as the logistic loss $\psi(z) = \log(1 + \exp(-z))$ and the square loss $\psi(z) = (1 - z)^2/4$. In particular, we can obtain a more quantitative results for the square loss.

**Theorem 2.** *Under the same condition of Theorem 1, when using $\psi(z) = (1-z)^2/4$ as the surrogate loss function, we have $R(f) - R^* \leq \sqrt{2\big(R_{LAC}(f_1, \ldots, f_K, f_{\mathsf{ac}}) - R^*_{LAC}\big)}$.*

Theorem 2 shows that the excess risk of $R_{LAC}$ upper bounds that of 0-1 loss. Thus, by minimizing the LAC risk $R_{LAC}$, we can obtain well-behaved classifiers on the testing distribution w.r.t. 0-1 loss.

**Remark 2.** Theorems 1 and 2 show the consistency for $R_{LAC}$ under class shift condition. Similar results can be easily obtained for $R^{shift}_{LAC}$ with the generalized class shift condition, due to the equivalence of $R^{shift}_{LAC}$ and the OVR risk, even when prior of known classes have changed.

**Finite-sample generalization error bound.** We establish the generalization error bound for the proposed approach in this part. Since the approach actually minimizes the empirical risk estimator $\widehat{R}_{LAC}$ with a regularization term of the RKHS $\mathbb{F}$, it is equivalent to investigate the generalization ability of classifiers $f_1, \ldots, f_K, f_{\mathsf{ac}}$ in the kernel-based hypothesis set $\mathcal{F} = \{\mathbf{x} \mapsto \langle \mathbf{w}, \Phi(\mathbf{x}) \rangle \mid \|\mathbf{w}\|_{\mathbb{F}} \leq \Lambda\}$, where $\Phi : \mathbf{x} \mapsto \mathbb{F}$ is a feature mapping associated with the positive definite symmetric kernel $\kappa$, and $\mathbf{w}$ is an element in the RKHS $\mathbb{F}$. We have the following generalization error bound.

**Theorem 3.** *Assume that $\kappa(\mathbf{x}, \mathbf{x}) \leq r^2$ holds for all $\mathbf{x} \in \mathcal{X}$ and the surrogate loss function $\psi$ is bounded by $B_\psi \geq 0$ and is $L$-Lipschitz continuous.[2] Then, for any $\delta > 0$, with probability at least $1 - \delta$ over the draw of labeled samples $D_L$ of size $n_l$ from the distribution of known classes $P_{kc}$ and unlabeled samples $D_U$ of size $n_u$ from $p^{te}_X(\mathbf{x})$, the following holds for all $f_1, \ldots, f_K, f_{\mathsf{ac}} \in \mathcal{F}$,*

$$R_{LAC}(f_1, \ldots, f_K, f_{\mathsf{ac}}) - \widehat{R}_{LAC}(f_1, \ldots, f_K, f_{\mathsf{ac}})$$
$$\leq \frac{2(K+1)\Lambda r}{\sqrt{n_l}} + 6\Lambda r \sqrt{\frac{2\log(4/\delta)}{n_l}} + \frac{2(K+1)L\Lambda r}{\sqrt{n_u}} + 3(K+1)B_\psi \sqrt{\frac{\log(4/\delta)}{n_u}}.$$

Based on Theorem 3, by the standard argument [30, 31], we can obtain the estimation error bound.

**Theorem 4.** *Under the same assumptions of Theorem 3 and let $\widehat{f}_1, \ldots, \widehat{f}_K, \widehat{f}_{\mathsf{ac}}$ be the optimal solution of the optimization problem (8) with certain $\lambda > 0$, with high probability, we have*

$$R_{LAC}(\widehat{f}_1, \ldots, \widehat{f}_K, \widehat{f}_{\mathsf{ac}}) - \inf_{\boldsymbol{f} \in \mathscr{F}} R_{LAC}(f_1, \ldots, f_K, f_{\mathsf{ac}}) \leq \mathcal{O}\left(\frac{K+1}{\sqrt{n_l}} + \frac{K+1}{\sqrt{n_u}}\right),$$

*where $\boldsymbol{f}$ denotes $(f_1, \ldots, f_K, f_{\mathsf{ac}})$ and $\mathscr{F} = \{\boldsymbol{f} \mid f_1, \ldots, f_K, f_{\mathsf{ac}} \in \mathbb{F}, \sum_{k=1}^K \|f_k\|^2_{\mathbb{F}} + \|f_{\mathsf{ac}}\|^2_{\mathbb{F}} \leq c^2_\lambda\}$. The parameter $c_\lambda > 0$ is a constant related to $\lambda$ in (8). We use the $\mathcal{O}$-notation to keep the dependence on $n_u$, $n_l$ and $K$ only, where the full expression can be found in the full paper.*

**Remark 3.** Theorem 3 and Theorem 4 show that, the estimation error of the trained classifiers decreases with a growing number of labeled and *unlabeled* data, which theoretically justifies the effecacy of our approach in exploiting unlabeled data. Experiments also validate the same tendency.

**Overview of theoretical results.** Recall that the goal of the LAC problem is to obtain classifiers that approach Bayes rule over the testing distribution, so we need to minimize the excess risk $R\big(\arg\max_{k \in \{1, \ldots, K, \mathsf{ac}\}} f_k\big) - R^*$. According to the consistency guarantee presented in Theorem 1, it suffices to minimize the excess risk $R_{LAC}(\boldsymbol{f}) - R^*_{LAC}$, which can be further decomposed into the estimation error and the approximation error as follows,

$$R_{LAC}(\boldsymbol{f}) - R^*_{LAC} = \underbrace{R_{LAC}(\boldsymbol{f}) - \inf_{\boldsymbol{f} \in \mathscr{F}} R_{LAC}(\boldsymbol{f})}_{\text{estimation error}} + \underbrace{\inf_{\boldsymbol{f} \in \mathscr{F}} R_{LAC}(\boldsymbol{f}) - R^*_{LAC}}_{\text{approximation error}}.$$

Theorem 4 shows that with an increasing number of labeled and unlabeled data, the excess risk converges to the irreducible approximation error, which measures how well the hypothesis set approximates the Bayes rule and is generally not accessible for learning algorithms [31]. Thus, the consistency and excess risk bounds theoretically justify the effectiveness of our approach.

## 5 Related Work and Discussion

This section discusses several research topics and techniques that are related to our approach.

**Class-incremental learning** [32] aims to handle new classes appearing in the learning process, and learning with augmented classes is one of its core tasks. Some early studies [6, 33] try to exploit unlabeled data for handling the LAC problem. Our approach differs from theirs as we depict the connection between known and augmented classes by the class shift condition rather than the geometric assumption, which leads to more clear theoretical understandings and better performance. Apart from the batch setting, researchers also manage to handle even more challenging scenario where augmented classes emerge in the streaming data [34, 35, 36, 37]. It is interesting to study that whether our approach can be tailored for the streaming setting.

**Open set recognition** [12, 38] is a cousin of the LAC problem studies in the computer vision and pattern recognition communities. As we have mentioned, several techniques or concepts are employed to depict the relationship between known and augmented classes, including open space risk [12, 13], nearest neighbor approach [14], extreme value theory [15] and the adversarial sample generation framework [39], etc. We note that many works in OSR implicitly use the feature semantic information to help identifying augmented classes. By contrast, our paper works on a general setting without such domain knowledge on the semantic information.

Although the approaches achieve nice empirical behavior and are underpinned by formal definitions or theories, their generalization error over testing distribution are less explored. Exceptions are works [16, 17, 18]. Authors focus on the Neyman-Pearson (NP) classification problem, where false positive ratio on known classes are minimized with the constraint on desired novelty detection ratio, or vice. Scott and Blanchard [16] and Blanchard et al. [17] provide one-side generalization bounds for both the novelty detection ratio and false positive ratio. However, the results mainly focus on the binary NP classification problem. The generalization error and excess risk analysis for the LAC problem, where multiple classes appear, is not investigated. Liu et al. [18] design a general meta-algorithm that can take any existing novelty detection approach as a subroutine to recognize augmented classes. They contribute to the PAC-style guarantee for the meta-algorithm on the novelty detection ratio, while performance on the false positive rate is less explored.

**Learning with positive and unlabeled examples** (LPUE), also known as PU learning, is a special semi-supervised learning task aiming to train a classifier for the binary classification with the positive and unlabeled data only [40, 27, 41, 42, 22]. One research line of LPUE is to exploit the risk rewriting technique to establish unbiased estimators for classifier training, which have also been adopted in our paper. The LAC problem with unlabeled data can be seen as a generalized LPUE problem by taking the known classes as positive. However, most studies on LPUE mainly focus on the binary scenario and established approaches are no longer unbiased in the multiclass case. For the multiclass scenario, Xu et al. [43] exploit the risk rewriting technique to train linear classifiers, which has also been adopted by Tsuchiya et al. [44] for ordinal regression. Although sharing similarity with [43, 44], our LAC risk is established in a quite different context and brings novel understandings for the LAC, through which more complex changing environments could be handled. Besides, the LAC risk allows more flexible implementations where the kernel method and deep model are applicable.

## 6 Experiments

We examine three aspects of the proposed EULAC approach: (Q1) performance of classifying known classes and identifying augmented classes; (Q2) accuracy of estimating mixture prior $\theta$ and its influence on EULAC; (Q3) capability of handling the complex changing environments (augmented class appears and prior of known classes shifts simultaneously). We answer the questions in following three subsections. In all experiments, classifiers are trained with labeled and unlabeled data, and are evaluated with an additional testing dataset which is never observed in training.

Table 1: Macro-F1 scores on benchmark datasets. The best method is emphasized in bold. Besides, ● indicates that EULAC is significantly better than others (paired $t$-tests at 5% significance level).

| Dataset | OVR-SVM | W-SVM | OSNN | EVM | LACU-SVM | PAC-iForest | EULAC |
|---|---|---|---|---|---|---|---|
| usps | $75.42 \pm 4.87$ ● | $79.77 \pm 4.97$ ● | $63.14 \pm 8.91$ ● | $61.14 \pm 6.27$ ● | $69.20 \pm 8.34$ ● | $55.69 \pm 13.3$ ● | $\mathbf{86.52 \pm 2.72}$ |
| segment | $71.78 \pm 5.12$ ● | $80.82 \pm 9.38$ ● | $85.10 \pm 5.98$ | $82.13 \pm 5.88$ ● | $40.69 \pm 12.5$ ● | $63.64 \pm 13.1$ ● | $\mathbf{86.17 \pm 5.80}$ |
| satimage | $54.67 \pm 9.80$ ● | $76.29 \pm 13.2$ ● | $62.48 \pm 11.2$ ● | $72.10 \pm 8.16$ ● | $51.56 \pm 17.3$ ● | $71.65 \pm 7.79$ ● | $\mathbf{81.25 \pm 6.18}$ |
| optdigits | $80.11 \pm 3.80$ ● | $87.82 \pm 4.64$ ● | $86.97 \pm 3.79$ ● | $72.00 \pm 8.33$ ● | $80.92 \pm 3.68$ ● | $71.65 \pm 5.46$ ● | $\mathbf{91.54 \pm 2.95}$ |
| pendigits | $72.78 \pm 5.19$ ● | $87.79 \pm 3.95$ | $86.69 \pm 3.39$ ● | $\mathbf{89.94 \pm 1.30}$ | $70.66 \pm 6.18$ ● | $73.21 \pm 4.52$ ● | $88.41 \pm 4.81$ |
| SenseVeh | $48.07 \pm 3.80$ ● | $45.96 \pm 2.32$ ● | $49.91 \pm 6.88$ ● | $51.24 \pm 3.91$ ● | $51.61 \pm 3.31$ ● | $54.12 \pm 7.19$ ● | $\mathbf{77.33 \pm 2.17}$ |
| landset | $60.43 \pm 7.65$ ● | $68.91 \pm 17.0$ ● | $73.25 \pm 9.23$ ● | $76.00 \pm 7.79$ ● | $53.59 \pm 9.88$ ● | $70.50 \pm 7.16$ ● | $\mathbf{85.70 \pm 4.46}$ |
| mnist | $66.74 \pm 2.76$ ● | $75.38 \pm 4.62$ ● | $57.75 \pm 10.9$ ● | $58.39 \pm 5.94$ ● | $63.53 \pm 7.58$ ● | $48.31 \pm 9.62$ ● | $\mathbf{80.66 \pm 5.38}$ |
| shuttle | $37.39 \pm 14.1$ ● | $58.48 \pm 34.5$ ● | $48.21 \pm 16.4$ ● | – | $34.18 \pm 13.4$ ● | $29.36 \pm 8.70$ ● | $\mathbf{66.49 \pm 17.9}$ |
| EULAC w/ t/ l | 9/ 0/ 0 | 8/ 1/ 0 | 8/ 1/ 0 | 8/ 1/ 0 | 9/ 0/ 0 | 9/ 0/ 0 | rank first 8/ 9 |

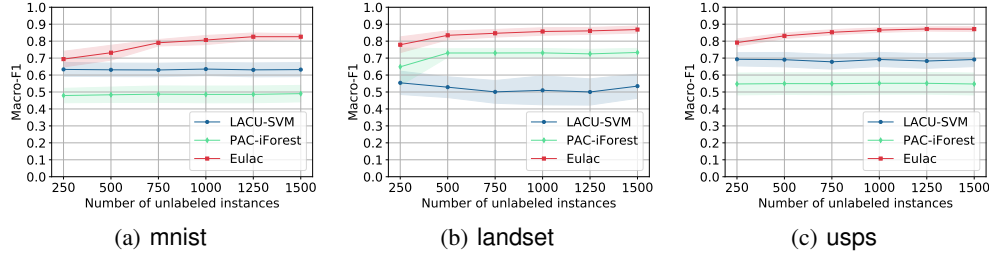

Figure 2: Macro-F1 score comparisons when the number of unlabeled data increases.

## 6.1 Performance Comparison

To answer Q1, we compare two implementations of EULAC (RKHS-based and DNN-based versions) with contenders on several benchmark datasets for various tasks. The overall performance over testing distribution is measured by the Macro-F1 score and accuracy. Meanwhile, we report AUC of augmented class score to evaluate the ability of identifying augmented classes. Due to space constraints, we provide detailed descriptions of datasets, contenders and measures in the full paper [25].

**Comparison on RKHS-based Eulac.** We adopt 9 datasets, where half of the total classes are randomly selected as augmented classes for 10 times. In each dataset, the labeled, unlabeled and testing data contain 500, 1000 and 1000 instance respectively. The instance sampling procedure repeats 10 times. Meanwhile, there are six contenders, including four without exploiting unlabeled data (OVR-SVM, W-SVM [13], OSNN [14], EVM [15]) and two using them (LACU-SVM [6], PAC-iForest [11]). Table 1 reports performance in terms of Macro-F1 score. Similar results for accuracy and AUC are shown in the full paper. We can see that EULAC outperforms others in most datasets. Note that it is surprising that W-SVM and EVM achieve better results than LACU-SVM and PAC-iForest, which are fed with unlabeled data. The reason might be that these methods require to set parameters empirically and the default one may not be proper for all datasets. By contrast, our proposed EULAC can perform *unbiased* cross validation to select proper parameters.

**Influence on the size of unlabeled data.** We vary the size of unlabeled data from 250 to 1500 with an interval of 250 on 3 datasets: mnist, landset, and usps. LACU-SVM and PAC-iForest are included for comparison. Figure 2 presents the Macro-F1 score and shows that the score of LACU-SVM remains unchanged or even drops in the three datasets, while performance of our approach is enhanced when provided with more unlabeled data, which is consistent with theoretical analysis in Section 4. This again validates that our approach can exploit unlabeled data effectively. Notice that PAC-iForest also enjoys sound theoretical guarantees, yet the guarantees only hold for the novelty detection ratio and thus the overall performance on the testing distribution is not promised to be improved.

**Comparison on deep models.** We also evaluate DNN-based EULAC, where the sigmoid loss $\psi(z) = 1/(1 + \exp(z))$ is used for the non-negative risk. The experiments are conducted on mnist, SVHN and Cifar-10 datasets, where six of all ten classes are randomly selected as known while the rest four are treated as augmented. The contenders are SoftMax, OpenMax [45], G-OpenMax [46], OSRCI [47]. All methods are trained based on the standard training split. The unlabeled data are sampled from part of the standard testing split and the rest instances are used for evaluation. Following the previous study [47], we report AUC of the augmented class in Table 2,

Table 2: AUC for DNN-based EULAC

| Methods | mnist | Cifar-10 | SVHN |
|---|---|---|---|
| SoftMax | $97.8 \pm 0.6$ | $67.7 \pm 3.8$ | $88.6 \pm 1.4$ |
| OpenMax | $98.1 \pm 0.5$ | $69.5 \pm 4.4$ | $89.4 \pm 1.3$ |
| G-OpenMax | $98.4 \pm 0.5$ | $67.5 \pm 4.4$ | $89.6 \pm 1.7$ |
| OSRCI | $\mathbf{98.8 \pm 0.4}$ | $69.9 \pm 3.8$ | $91.0 \pm 1.0$ |
| EULAC | $98.6 \pm 0.4$ | $\mathbf{85.2 \pm 2.0}$ | $\mathbf{91.2 \pm 2.8}$ |

and results of contenders are also from [47]. DNN-based EULAC can learn nice detection score for identifying augmented classes, which validates its efficacy.

## 6.2 Issue of Mixture Proportion

To answer Q2, we conduct experiments on mnist dataset, where the true mixture proportion varies from 0.1 to 0.9. Other configurations are the same as those in Section 6.1.

**Influence and accuracy on the estimation of $\theta$.** Figure 3 plots the sensitivity curve, where the estimated prior $\widehat{\theta}$ varies from 0.1 to 0.9 under different ground-truth mixture proportions $\theta$. We observe that a misspecified mixture proportion will clearly lead to performance degeneration. Interestingly, the degeneration is not isotropy—a larger misspecified value would be much more benign than a smaller one. We mark averaged estimated values of KME-base (♦) and KME-shift (★). Evidently, the estimator gives high-quality estimated prior $\widehat{\theta}$, close to the ground-truth value, which prevents our approach from performance degeneration.

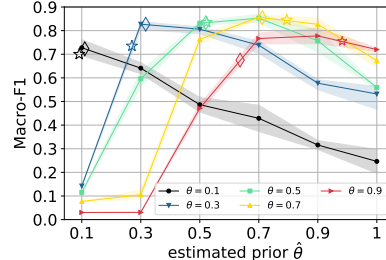

Figure 3: Influence and estimation accuracy of mixture proportion $\theta$.

## 6.3 Handling Complex Changing Environments

To answer Q3, we compare our approach with several baselines when augmented classes appear and prior of known classes shifts simultaneously. The experiments are simulated on mnist dataset, where classes $\{1, 3, 5, 7, 9\}$ are known and share the equal prior in labeled data. $\{2, 4, 6, 8, 10\}$ are taken as the augmented classes and account for $50\%$ in unlabeled data. As for the prior shift in the testing distribution, we scale the the prior of five known classes to $[1 - \alpha, 1 - \alpha/2, 1, 1 + \alpha/2, 1 + \alpha] \times 0.2$ respectively, where parameter $\alpha$ controls shift intensity ranging from 0 to 0.7.

**Contenders.** Contenders include LACU-SVM, OVR-shift and three variants of EULAC (EULAC-base, EULAC-base++ and EULAC-shift), where LACU-SVM and EULAC-base do not consider the shift on known classes' prior, while OVR-shift and EULAC-base++ take it into account but are biased. EULAC-shift is the unbiased estimator. For all approaches, class prior $\theta_{te}^k$ is estimated by KME-shift. Detailed descriptions of contenders can be found in the full paper [25].

**Results.** Since Macro-F1 is an insensitive measure for the prior shift scenario, we report the accuracy for contenders in Figure 4. First, with the increase of shift intensity, methods without considering prior shift (LACU-SVM, EULAC-base) suffer from marked performance degeneration, which shows the importance for handling distribution change of known classes with augmented classes. Besides, EULAC-shift achieves the best accuracy with high shift intensity and retains comparable performance with its baselines when there is no prior shift. The results validate the efficacy and safety of our proposal in complex environments.

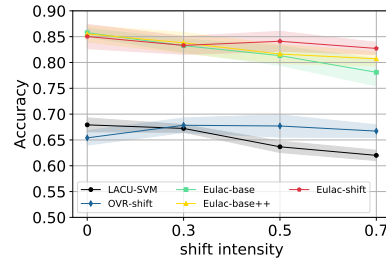

Figure 4: Comparison in complex environments (augmented classes & prior shift).

## 7 Conclusion

In this paper, we investigate the problem of learning with unobserved augmented classes by exploiting unlabeled training data. We introduce the *class shift condition* to connect known and augmented classes, based on which an unbiased risk estimator can be established. By empirically minimizing the risk estimator with various hypothesis sets, we design the EULAC approach, supported by both consistency and generalization error analysis. Moreover, with the generalized class shift condition, we show the potential of our approach for handling a more general setting of class distribution change, where augmented classes appear and the prior of known classes shifts simultaneously. Extensive empirical studies confirm the effectiveness of the proposed approach. In the future, we will investigate whether our approach can be tailored for the streaming setting. Besides, it is also interesting to consider even more general scenarios of class distribution change than the problem settings studied in this paper, in order to handle more realistic changing environments.

## Acknowledgements

This research was supported by the NSFC (61751306, 61921006) and the Collaborative Innovation Center of Novel Software Technology and Industrialization. Meanwhile, the authors want to thank Yu-Hu Yan for reading the draft and the anonymous reviewers for the helpful and insightful comments.

## Broader Impact

In this paper, we develop the EULAC, an approach exploiting unlabeled data for learning with augmented classes. The augmented classes appear in many applications, such as unobserved animals appear in species recognition task [1] and unexpected background images exist in object detection [12]. Our approach offers a way to improve the robustness of the learning system for these applications by identifying the unseen augmented classes more accurately. Nevertheless, we also admit it would raise concerns when applying these techniques to some malicious applications. For example, one could employ ML systems to detect rare animals, resulting in an increased probability of rare animals being hunted and thus making the animals even more dangerous. Therefore, we should call for laws and regulations to limits the use of ML techniques in such applications.

On the other hand, it is also crucial to facilitate learning systems with the capability of tackling the augmented classes. Many applications require such robustness and will benefit from our techniques, and the potential risk is believed to be manageable with more sound human regulations.

## Footnotes

[1] We redefine all the distributions over the space $\mathcal{X} \times \mathcal{Y}$, where $p_{XY}^{\mathsf{kc}}(\mathbf{x}, \mathsf{ac}) = 0$ for all $\mathbf{x} \in \mathcal{X}$

[2]Common surrogate loss functions satisfy these conditions, such as logistic loss, exp loss and square loss.

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
