[Reviews · NeurIPS 2020]

Review 1

Summary and Contributions: The paper proposes a novel approach to solving problems with in the setting of learning with augmented classes and unlabeled data (LACU). The authors re-frame the problem and derive a formulation in which they can minimize the (unknown) risk function efficiently if they know the class proportions in the test distribution.

Strengths: The main contributions are: * Prop 1 & 2: characterization of the full OVR risk if labels for new class were given, showing that it can be recovered from labeled train and unsupervised test data (if true mixture proportion is known) * Theoretical analysis of proposed framework & kernel algorithm: consistency & finite sample bounds (Thms 1-4) Other contrib: * formalization of class-shift & generalized class-shift condition * simple derivation of kernelized & dl-based algorithms to optimize the introduced objectives * empirical evaluation of proposed methods

Weaknesses: - Relative basic theory - hard to compare empirical results to literature

Correctness: yes

Clarity: yes

Relation to Prior Work: yes

Reproducibility: Yes

Additional Feedback: The proposed methodology is simple, straightforward and elegant. All theoretical results are relatively basic, either simple derivations from [16] or they follow from standard learning theory results. (which of course need not be a problem) The empirical evaluation looks sufficiently thorough. However, regarding the bad results for the most relevant contenders, LACU-SVM & PAC-iForest: it would have been useful to include data sets from the corresponding papers (eg 20newsgroup for LACU-SVM) for comparison and test whether you're able to reproduce their reported results - like it is, it is hard to compare the empirical performance properly. The paper is generally well written and very easy to understand. Typos & minor comments: Table 1: significance level would be 5%, not 95% "high-stake" not "high-stack" in line 14 line 91: classifier' index 1,...,K+1 or 1,...,K,nc is somewhat inconsistently used - maybe better use one of the two throughout? (such as between eq (2) and Prop 1) line 135 is somewhat sloppy - it's clear what you mean, but theta_tr^i were no-where defined, and if they actually where the class priors in the training data, then the theta from eq (1) would be 0 line 137 - some word is missing, maybe an "if"? line 168 - "empirical", not "empirically" line 198-199 - missing "that" and "the" "I looked at the author rebuttal, but this does not change my stance about the paper. Hence, I keep my current score."


Review 2

Summary and Contributions: This paper proposes an unbiased risk estimator for learning with the augmented class if the unlabeled data that contains augmented classes are provided. The trick to derive an unbiased estimator is based on the risk rewriting technique, which has been used extensively in weakly-supervised classification, e.g., positive-unlabeled classification, and complementary-label classification. Theoretical analyses and experiments are provided to illustrate the effectiveness of the proposed method.

Strengths: Theoretical work is limited in this line of research (learning with augmented class). This paper provides theoretical results for learning under this scenario which is quite insightful and quite unique in this line of research.

Weaknesses: I do not see a critical weakness overall. 1. It is useful to add weaknesses/limitations of the proposed method. For example, my guess is that when the number of classes is large, this approach may be inappropriate because mixture proportion estimation can be unreliable. 2. Do we need to commit ourselves to the OVR loss? It may be a bit restrictive to focus on the OVR loss. It seems to me that we can straightforwardly consider a general infinite-sample consistent multiclass surrogate loss. If the authors can clarify how difficult it may occur when considering a loss function such as softmax cross-entropy loss apart from the convex formulation may not be obtained, I may consider increasing the score.

Correctness: I think the proposed method is sound.

Clarity: The paper is well-written overall and easy to understand. Minor comments: line 247 and Figure 2&4: Eulac -> EULAC

Relation to Prior Work: I think it is important to point out that the proposed risk estimator is very similar if not identical to that of [1] and [2] although they are used in the different context. In [2], they also discussed the convex formulation based on the loss that satisfies the linear-odd condition like this paper. Having said that, using the proposed method for learning from augmented class in this paper is still novel and highly relevant and useful. Moreover, it can be insightful to interpret the relationship of the proposed method with positive-unlabeled classification, which several papers are already cited in the paper. In this context, positive means seen classes and negative is an unseen class [1]. [1] Xu et al., Multi-Positive and Unlabeled Learning, IJCAI 2017 [2] Tsuchiya et al., Semi-Supervised Ordinal Regression Based on Empirical Risk Minimization, ArXiv 2019

Reproducibility: Yes

Additional Feedback: 1. How to use the non-negative risk estimator in this problem? In particular, where to add the max-operator in the risk estimator? I think it is important to clarify this part. 2. In experiments, where the sigmoid loss is used for the deep model. I am aware that the choice of loss is identical to Kiryo et al. My question is have you tried different loss functions? if so, how was their performance compared with the sigmoid loss? 3. For the analysis of infinite-sample consistency in Theorem 1, loss function choice is quite restrictive and does not cover many losses such as the sigmoid loss. In my opinion, R_{LAC} is basically identical to R_{te}. Therefore, the minimizer of R_{LAC} must be identical to R_{te} regardless of a loss function choice. Thus, it seems trivial to me that we can get a bayes classifier by minimizing R_{LAC}. Having said that, do we need to prove Theorem 1 for this problem? Theorem 1 says the same thing but is limited to a restricted choice of losses. Moreover, as far as I understand, the "infinite-sample consistency" or "classification-calibration" in the seminar work of Zhang mentioned in this paper is a property of a "individual loss function" not the "risk". Thus, saying a risk is "infinite-sample consistent" may be strange (please correct me if I am wrong). ===================== I have read the other reviews and the author feedback. I appreciate the authors for addressing my questions and concerns. 1. On Theorem 1: Thanks for the clarification on the necessity of Theorem 1. 2. On the extension to a general loss function: Thanks. It is good to know that this approach can be straightforwardly extend to a general loss function with no difficulties. After reading the author's response and other reviews, I think this paper has merits that outweigh flaws and would like to stand by my assessment (6/10). A big challenge is that the assumption may sound unrealistic for this problem (identical class-conditional probabilities even when a new class is added, only the class prior changes). But without this, it would be difficult to derive theoretical guarantee. Although this risk rewriting technique has been used extensively in several previous works, knowing that this approach can be useful for this line of research is still beneficial for the community. More minor comments: 1. There are the usages of "mnist" and "MNIST" in this paper. If there is no important reason if would be better to unify it. 2. The word "testing" distribution might be confusing whether the test data is used when training a classifier. How about using the word "target" distribution instead (following the domain adaptation literature)? 3. Lines 273-274: "The unlabeled data for EULAC are sampled from the testing data and the rest are used for evaluation": this explanation might cause confusion whether the test data is used when training a classifier as well. Maybe it is nice to emphasize that the test data for evaluation is never observed by a classifier in the training phase.


Review 3

Summary and Contributions: (Post-rebuttal) I have kept my rating the same post-rebuttal. I do not agree with the rejection recommendation from reviewer #4, who is mischaracterizing the evaluation regime used in the paper. I believe this is a misunderstanding coming from a legitimate weakness in the paper: it was unclear when, exactly, in operation the unbiased risk estimator is established. Related to this is the practical question of where, exactly, the unlabeled dataset comes from. The rebuttal does clear some of the confusion up for me, but the paper needs to be more clear on these points. This paper studies the problem of open set recognition, which it terms “Learning with Augmented Classes.” In this problem, new classes that do not appear in the training data might emerge in the testing phase. The paper argues that much of the existing work focuses on exploring the geometric properties of new classes, which can lead to good empirical performance but lacks a theoretical understanding – especially for generalization in testing. The key idea of this paper is to exploit unlabeled data to approximate the distribution of new classes, establishing an unbiased risk estimator for the testing distribution. It is claimed that the proposed algorithm can adapt to complex changing environments where new classes come and the prior of known lasses changes simultaneously. A series of experiments is provided that demonstrates feasibility for the proposed approach.

Strengths: Positives: + I really like the key idea of this paper: exploiting unlabeled data to approximate the distribution of new classes that appear in open set recognition scenarios. The use of unlabeled data for this purpose has not been explored to any great extent in prior work, and this paper provides a convincing path forward. + The paper provides a theoretically grounded convex RKHS-based approach, as well as a non-convex neural network approach embodying the same general idea. + The empirical analysis shows good results for the proposed approach (although I have a question about the comparison experiments for the RKHS-based Eulac). + The paper, for the most part, is well written, and combines formal analysis with intuitive explanations. (There are a few places where the clarity of writing can be improved though; I point these out below.)

Weaknesses: Negatives: - The paper uses non-standard terminology for a well-known problem in the literature. - Some of the claims made about prior work are not accurate. - The paper is unclear on when, exactly, in operation the unbiased risk estimator established. Related to this is the practical question of where, exactly, the unlabeled dataset comes from. - There is some question in my mind as to whether or not the comparison experiment reflected in Table 1 is fair, based on the way hyperparameters are set.

Correctness: Some of the claims made about prior work are not accurate. The empirical methodology may need improvement. (See my detailed comments below.)

Clarity: The paper, for the most part, is well written, and combines formal analysis with intuitive explanations. (There are a few places where the clarity of writing can be improved though; I point these out below.)

Relation to Prior Work: The paper cites relevant related references, but is not correct in some of the claims made about the differences between the proposed method and those that have been published previously.

Reproducibility: Yes

Additional Feedback: Detailed Comments: Overall, the idea of exploiting unlabeled data to approximate the distribution of new classes at test time is a good one. Prior work [9,10,12] has looked at using labeled “known unknown” data in a supervised context to support generalization, but this requires some human intervention to label the data as not being from the known classes of interest. Further, those data are not sampled from the testing distribution (since this process happens at training time). The proposed approach gives a more flexible path forward for estimating the distribution of unknowns (and knowns) from the testing distribution. A major problem in the presentation of this work is that it introduces new terminology for a well-known problem in the literature. “Open Set Recognition” is commonly used to describe the same exact problem dubbed “Learning with Augmented Classes” here. There is absolutely no need to change the name of this problem. In fact, this work will have more impact if it is properly associated with the area of research it describes. Line 30 of the Supp. Mat. even acknowledges that these two terms are synonymous. Line 30 states: “Although various algorithms are proposed with satisfactory empirical performance, their theoretical properties are generally unclear.” This is not true for some of the references cited in the introduction. [9] proposes a theory for open set recognition underpinned by a formal definition of open space risk. [10] and [12] describe algorithms that are underpinned by the formal extreme value theory. A big question I have with respect to this work is at what point is the unbiased risk estimator established? Section 2.1 states that “In our setup, the learner can additionally receive an unlabeled dataset… sampled from the testing distribution.” However, in Sec. 2.2, the paper tells us that “the testing distribution is unavailable in the training stage due to the absence of new classes. Fortunately, we show that given mixture proportion $\theta$, the testing distribution can be approximated with the labeled and unlabeled data.” I could see this happening in two ways: (1) known labeled data is curated at training time, and unknown data is sampled from the world the model will be used in at testing time, and (2) known labeled data is curated at training time, and “known unknown” data is sampled from an available source, also at training time. Which way is used in practice? A related question is what happens when multiple class shifts occur over time (i.e., the environment changes, which causes a change in the testing distribution)? Is it possible to relearn the unbiased risk estimator in some efficient way? (Something akin to incremental learning.) The final sentence of the paper seems to indicate that this might still be on the TODO list: “In the future, we will take emerging new classes as well as other types of distribution changes into account, in order to handle more realistic changing environments.” RKHS vs. Neural Network Implementation: What is the practical tradeoff between the convex and non-convex formulations? Do we have to worry about non-convexity for open set recognition? I have a concern over how the comparison experiment for the RKHS-based Eulac was conducted. The paper states the following in Sec. 5: “Note that it is surprising that W-SVM and EVM achieve better results than LACU-SVM and PAC-iForest, which are fed with unlabeled data. The reason might be that these methods require to set parameters empirically and the default one may not be proper for all datasets. By contrast, our proposed EULAC can perform unbiased cross-validation to select proper parameters.” The Supp. Mat. seems to indicate that most of the comparison approaches used default values for hyperparameters. But according to the description in the paper, the Eulac method gets to choose its parameters via cross-validation. The same could be true of the comparison approaches if a hyperparameter optimization method is used to tune any free parameters. Some of the comparison approaches were not developed with the full complement of datasets presented in this work, thus there isn’t a strong expectation for their default settings to be the best here. Minor Problems: The caption of Figure 1 is very terse. A more descriptive caption for this figure would be helpful. The broader impact statement doesn’t really get at the potential problems of failure modes in the proposed approach. Typos: Line 11: “and prior of known classes” should be “and the prior of known classes” Line 61: “motivates us the design” should be “motivates the design” Line 168: “empirically observations” should be “empirical observations”


Review 4

Summary and Contributions: This paper proposes a novel unbiased risk estimator for the problem of learning with augmented classes (LAC). With the novel unbiased risk estimator, a novel LAC algorithm is developed. The theoretical effectiveness of the method has been proved by both consistency and generalization error analysis. According to the experimental results providing by the paper, the proposed method outperforms several state-of-the-art methods. The paper is globally well organized. But there are some major problems. See Weaknesses.

Strengths: 1. The paper is clearly written and easy to follow. 2. Detailed theoretical analysis and extensive experimental comparison are given.

Weaknesses: 1. The assumption that the testing distribution were available seems unrealistic. In machine learning, test set is used to provide an unbiased evaluation of a final model fit on the training dataset, especially the generalization ability. If we assume that the testing distribution were available, then the model will be oriented to fit the test set. In this situation, the test set cannot give unbiased evaluation. The authors should give more explanation for the assumption and show that the assumption can be used in practice. 2. The novelty of this paper is limited. The construction of the proposed unbiased risk estimator is a common technology widely used in lots of learning settings, for example in the Positive Unlabeled Learning. 3. In Theorem 4, ‘…with high probability…”is not clear. The probability should be quantitatively. 4. In the experiment, comparison on deep models, the paper proposes that ‘The unlabeled data for EULAC are sampled from the testing data and the rest are used for evaluation’. What are the testing data used in other method? All the testing data or same with EULAC? Overall, this paper shows some math skills, but does not slove any real problems in practice and does not conduct strong theoretical analysis. ------------ I read the rebuttal. In the rebuttal, the author cliams that " We do NOT use any testing sample in training time. Instead, we try to approximate the testing distribution by only labeled and unlabeled training data". Obviously, it is a big erroneous. The paper study the problem of learning with augmented classes (LAC), where new classes that do not appear in the training data might emerge in the testing phase. The basic assumption is that the testing data have new classes which is not appear in the training data. Based on this setting. The distribution of training data and testing data should be totally different. But the author use training data to approximate the testing distribution. It is contradictory and not correct. In the rebuttal, the author cliams that "we are actually studying the same setting as previous works [3,15]". I carefully checked reference [15]. However, I find the setting of [15] is totally different from this paper. [15] only requires unlabeled samples drawn from a mixture of nominal data distribution and unknown alien distribution, but the mixture proportion $\theta$ of the unlabeled samples is not required to be same with the test data. It means the distribution of test data can be different with the unlabeled data. However, in this paper, the mixture proportion $\theta$ is necessary, and the distribution of test data must be same with the distribution. In paper [15], it is presented that “At training time, we assume that $D_m$ is a mixture distribution, with probability α of generating an alien data point from Da and probability of $1 − \alpha$ of generating a nominal point. Our results hold even if the test queries come from a mixture with a different value of α as long as the alien test points are drawn from $D_a$.” In addition, my concern about Theorem 4 is that ‘…with high probability…”is not clear. The probability should be quantitatively defined. The author did not address my concern about the Theorem 4.

Correctness: It seems correct.

Clarity: Clear.

Relation to Prior Work: some important works are missing.

Reproducibility: No

Additional Feedback:

[Author Response · NeurIPS 2020]

**[To Reviewer #1]** Thanks for your appreciation of this work! We address your concern as follows.

**Q1:"...include data sets from corresponding papers (eg 20newsgroup for LACU-SVM)"** During rebuttal period, we

compared EULAC and LACU-SVM on 20newsgroup for 100 repeated experiments, and their win/tie/loss results are:

44/30/26 for linear kernel and 100/0/0 for Gaussian kernel. This clearly shows the advantage of our method. Note that

parameter tuning has been conducted for LACU-SVM, please refer to the response to Q4 of Reviewer #3 if interested.

**[To Reviewer #2]** Thanks for your detailed review and insightful comments. We answer your main questions as follows.

**Q1:"Do we need to commit ourselves to the OVR loss?...considering a loss function such as softmax cross entropy**

**loss apart from the convex formulation..."** You are absolutely correct! Proposition 1 (unbiased property) can be

generalized to arbitrary multiclass surrogate loss $\Phi(\boldsymbol{f}(\mathbf{x}), y)$. The derivation is straightforward by the condition

$(1 - \theta) \cdot P_{new} = P_{te} - \theta \cdot P_{tr}$ and we have unbiased risk estimator $R_{LAC} = \theta \cdot \mathbb{E}_{(\mathbf{x},y) \sim P_{tr}}[\Phi(\boldsymbol{f}(\mathbf{x}), y) - \Phi(\boldsymbol{f}(\mathbf{x}), \mathsf{nc})] +$

$\mathbb{E}_{\mathbf{x} \sim P_{te}^{X}}[\Phi(\boldsymbol{f}(\mathbf{x}), \mathsf{nc})]$. The reason for choosing OVR loss in our paper is to obtain a convex formulation and thus easy to

optimize. If convexity is not required (e.g., NN implementation), we can use more flexible multiclass loss and binary

loss with above $R_{LAC}$, so the mentioned cross entropy loss is also applicable. We will make this clear in the revision.

**Q2:"How to use the non-negative risk estimator in this problem?"** The max operator can be added for each binary

classifier to avoid the negative loss. We will add more elaborations about the formulation in the revision.

**Q3:"My question is have you tried different loss functions?"** Yes, we initially use softmax cross entropy loss for deep

models. However, it does not converge in experiments. So we instead use sigmoid loss following Kiryo et al. [24].

**Q4:The issue about Theorem 1.** Theorem 1 serves as a guide to choose binary loss for OVR scheme. Proposition 1

($R_{LAC} = R_{te}$) only ensures the unbiasedness for surrogate loss, which does not mean we can optimize $R_{LAC}$ over

surrogate loss to obtain a model performing well on 0-1 loss. Thus, a consistency guarantee (Theorem 1) is necessary.

**[To Reviewer #3]** Thanks for the detailed review and helpful comments. We address your main concerns as follows.

For the other minor issues, we will discuss in the paper and revise the paper according to your suggestions.

**Q1:About the terminology of LAC and OSR.** We would like to revise the terminology in the revision if it is allowed.

**Q2:"Some of the claims made about prior work are not accurate."** Thanks, previous works [9,10,12] indeed provide

theoretical analysis for OSR. Our primary claim is that we are the first to provide the generalization analysis in this line

of research. We will revise the paper and appropriately acknowledge theoretical contributions of previous works.

**Q3:"...what point is the unbiased risk estimator established?"** We use way (2). In addition to labeled training data,

there is a set of unlabeled data from an available source reflecting underlying environments. We can then establish

an unbiased risk estimator for the underlying distribution in testing time (where new classes emerge now). Moreover,

we note that our method can also be extended to *transductive* setting [aka, way (1)]: only labeled data are available in

training time. Then, we can use part of testing samples as unlabeled data to establish the unbiased estimator and employ

that to the rest (analog to cross-validation procedure). We will clarify the setting and add the remark in the revision.

**Q4:"...there isn't a strong expectation for their default settings**

**to be the best"** As argued by [3], LACU-SVM is not sensitive

to parameters, so we use default ones. In [15], PAC-iForest is

somehow sensitive to core parameter $q$, and we have tuned it

in experiments. During rebuttal period, we performed a case

study on mnist and validate their claim (see Figure 1 left). We

further conducted parameter tuning thoroughly for LACU-SVM (4

parameters) and PAC-iForest (6 parameters) on *each* dataset (see

Figure 1: case study (left); parameter study (right).

Figure 1 right). Optimal tuning results are slightly better than the default one, while they are still worse than our EULAC.

**[To Reviewer #4]** Thanks for your review. Below we would like to clarify several serious misunderstandings.

**Q1:"the testing distribution were available seems unrealistic."** This is a misunderstanding: We do NOT use any

testing sample in training time. Instead, we try to approximate the testing distribution by only labeled and unlabeled

training data. On the other hand, we are actually studying the *same* setting as previous works [3,15]. It is realistic that

in many tasks unlabeled data can be collected for learning with new classes. Examples are the insect recognition [15]:

there exist unobserved insects missed by labeled data, and unlabeled data could help to identify them.

**Q2:About the novelty.** We admit that rewriting the risk estimator has been used in several weakly supervised learning

problems, but our paper is the *first* to provide an unbiased risk estimator for learning with augmented classes, which is a

unique view in the line of research. By doing so, we can handle even more complex changing environments (like target

shift considered in our paper), and we can provide sound and clear theoretical guarantees for the proposed approach.

We hope the reviewer could check comments from other reviews. For example, Reviewer #1 said "The proposed

methodology is simple, straightforward and elegant."; Reviewer #2 said "This paper provides theoretical results for

learning under this scenario which is quite insightful and quite unique in this line of research."; Reviewer #3 said "The

use of unlabeled data for this purpose has not been explored to any great extent in prior work, and this paper provides a

convincing path forward." So we believe our work has sufficient novelty and contributions to the community.

**Q3:About the testing data.** All the methods share the *same* set of testing data in each experiment.

[Meta-Review · NeurIPS 2020]

Despite a disagreement from R4, there is a consensus among most of knowledgeable reviewers that this is a good paper. After reading the paper, I also concur that the problem considered in this paper is important and the proposed solution is interesting, novel, and simple. Hence, I recommend that the paper is accepted as a poster. This paper considers the problem of learning with augmented class and unlabeled sample (aka open set recognition). That is, the authors assume that at test time a new class which is not available at training time can emerge. As a result, there is a distributional shift between training distribution and test distribution (i.e., non-i.i.d. setting). This itself is an important problem. The idea proposed in this work is to use labeled training data together with "unlabeled" data from the test distribution (with augmented class) to construct an unbiased estimate of the risk from which the classifier can be learned. Although the risk rewriting technique has been used extensively in several previous works, I find the idea of using unlabeled data to construct an unbiased estimate of the risk quite interesting. From the reviews, R1, R2, R3 also appear to support the acceptance of this paper. Nevertheless, R4 raised a major concern during the discussion regarding the use of test data during training. While this is a valid point and R4 seems to take his/her stance after thorough discussion, I do not think it can be used to justify a rejection. In fact, I believe that the misunderstanding may stem from an ambiguity in the presentation of how the unlabeled data is used in the training process. Hence, I would like to suggest that the authors improve the presentation of the camera-ready version and clarify the role of unlabeled data in this framework (especially in the experiment section).